# Maternal Plasma RNA in First Trimester Nullipara for the Prediction of Spontaneous Preterm Birth ≤ 32 Weeks: Validation Study

**DOI:** 10.3390/biomedicines11041149

**Published:** 2023-04-11

**Authors:** Carl P. Weiner, Helen Zhou, Howard Cuckle, Argyro Syngelaki, Kypros H. Nicolaides, Mark L. Weiss, Yafeng Dong

**Affiliations:** 1Department of Obstetrics and Gynecology, University of Kansas Medical Center, Kansas City, KS 66160, USA; 2Department of Molecular and Integrative Physiology, University of Kansas Medical Center, Kansas City, KS 66160, USA; 3Rosetta Signaling Laboratory LLC, Phoenix, AZ 85018, USA; 4Faculty of Medicine, Tel Aviv University, Ramat Aviv, Tel Aviv 69978, Israel; 5Fetal Medicine Research Institute, King’s College Hospital, London SE5 9RS, UK; 6Departments of Anatomy and Physiology & Midwest Institute of Comparative Stem Cell Biology, Kansas State University, Manhattan, KS 66503, USA

**Keywords:** pregnancy, preterm birth, screening tests, plasma transcriptome

## Abstract

The first-trimester prediction of spontaneous preterm birth (sPTB) has been elusive, and current screening is heavily dependent on obstetric history. However, nullipara lack a relevant history and are at higher risk for spontaneous (s)PTB ≤ 32 weeks compared to multipara. No available objective first-trimester screening test has proven a fair predictor of sPTB ≤ 32 weeks. We questioned whether a panel of maternal plasma cell-free (PCF) RNAs (*PSME2*, *NAMPT*, *APOA1*, *APOA4*, and *Hsa-Let-7g*) previously validated at 16–20 weeks for the prediction of sPTB ≤ 32 weeks might be useful in first-trimester nullipara. Sixty (60) nulliparous women (40 with sPTB ≤ 32 weeks) who were free of comorbidities were randomly selected from the King’s College Fetal Medicine Research Institute biobank. Total PCF RNA was extracted and the expression of panel RNAs was quantitated by qRT-PCR. The analysis employed, primarily, multiple regression with the main outcome being the prediction of subsequent sPTB ≤ 32 weeks. The test performance was judged by the area under the curve (AUC) using a single threshold cut point with observed detection rates (DRs) at three fixed false positive rates (FPR). The mean gestation was 12.9 ± 0.5 weeks (range 12.0–14.1 weeks). Two RNAs were differentially expressed in women destined for sPTB ≤ 32 weeks: *APOA1* (*p* < 0.001) and *PSME2* (*p* = 0.05). *APOA1* testing at 11–14 weeks predicted sPTB ≤ 32 weeks with fair to good accuracy. The best predictive model generated an AUC of 0.79 (95% CI 0.66–0.91) with observed DRs of 41%, 61%, and 79% for FPRs of 10%, 20%, and 30%, including crown–rump length, maternal weight, race, tobacco use, and age.

## 1. Introduction

Preterm birth (PTB) is the leading cause of perinatal mortality and neonatal morbidity worldwide. Over 15 million PTBs annually lead to more than a million childhood deaths and lifelong sequelae for many survivors [1]. The global failure to reduce the rates of very (32 to 28 weeks) and extreme preterm (<28 weeks) birth reflects a poor understanding of the mechanisms leading to these births, paired with a limited ability to accurately identify women early in pregnancy who are at ‘high risk’ for PTB ≤ 32 weeks. Since the majority of prematurity-related costs are associated with these births [2,3], their accurate prediction and prevention would have maximal patient and societal impact.

Current screening for PTB is heavily dependent on obstetric history, but nullipara lack an obstetric history and are at higher risk for spontaneous (s)PTB ≤ 32 weeks compared to multipara [4]. No available screening test performed under 20 weeks has proven a fair predictor of sPTB ≤ 32 weeks. Smith [5] tested the predictive value of maternal characteristics and second-trimester alpha-fetoprotein and ß-human chorionic gonadotropin in 84,000 nulliparas. No model generated a receiver operating characteristic area under the curve (AUC) above 0.67. Van der Ven [6] tested the predictive ability of transvaginal sonographic cervical length (CL) at 16–21.9 weeks in 5700 nulliparas and 6200 low-risk multiparas. Though CL performed slightly better in nullipara, it remained a poor predictor (sPTB < 37 weeks, AUC 0.61 vs. 0.56; sPTB < 34 weeks, AUC 0.63 vs. 0.58), a finding supported by Rosenbloom [7]. Esplin [8] studied 9400 nulliparas at 16–22 weeks with both CL and cervical fetal fibronectin (fFn). No AUC exceeded 0.53 for sPTB < 37 weeks and 0.61 for sPTB < 32 weeks. While Esplin did not combine the markers at 16–22 weeks, they did at 22–30 weeks; there was no improvement in the accuracy compared to CL alone. Clearly, these tests are used by caregivers, not because of their accuracy, but because there is a testing void.

Prior reports demonstrate a maternal blood test consisting of five plasma cell-free (PCF) RNAs (*PSME2*, *NAMPT*, *APOA1*, *APOA4*, and *Hsa-Let-7g*) measured at 16–20 weeks may be a good predictor of sPTB ≤ 32 weeks [9,10]. A prospective cohort validation study revealed specific panel markers had good to excellent predictive accuracy for PTB ≤ 32 weeks, whether due to spontaneous labor (± preterm premature rupture of membranes (PPROM)) or early-onset preeclampsia (EOP) < 34 weeks independent of parity [10]. While the predictive accuracy for sPTB ≤ 32 weeks was encouraging (AUC = 0.83), antenatal screening has moved progressively to the first trimester for EOP and potentially sPTB ≤ 32 weeks. The objective of this study was to test the potential of the five RNA panel to predict sPTB ≤ 32 weeks in the first trimester nullipara.

## 2. Materials and Methods

In collaboration with the Fetal Medicine Research Institute, King’s College Hospital, London, UK, 60 first-trimester women prospectively enrolled in the Institute’s IRB-approved biobank for future research were randomly selected by one of the authors (AS) subject to the following conditions: nulliparous, healthy, and a pregnancy ending either at term or with sPTB ≤ 32 weeks. There were 20 controls and 40 cases. The racial distribution was the same for cases and controls: 50% self-described White and Black. Women who developed preeclampsia or had known comorbidities were excluded from this study. Gestational age was based on the first-trimester crown–rump length (CRL). 

### 2.1. Molecular Tests

RNA was extracted from 500 µL of EDTA plasma using a proprietary method (Rosetta Signaling Laboratory, Phoenix, AZ, USA). The mean total RNA extracted was 26.7 ± 9.5 µg/mL (median 18.7 µg/mL; range 8.0–76.3 µg/mL). RNA yield was assessed with a Nano spectrometer (NanoDrop Technologies, Wilmington, DE, USA) and integrity confirmed by the Agilent Bio-analyzer (Agilent, Santa Clara, CA, USA). RNA expression was quantified by qRT-PCR, as described previously [9]. All laboratory members were blind to pregnancy outcome.

### 2.2. qRT-PCR Assays

mRNA RT: RNA samples were diluted with a master mix including dNTP mix, Omniscript Reverse Transcriptase, and Random Primer (Invitrogen, Carlsbad, CA, USA). mRNA was converted into cDNA at 37 °C for 60 min, per manufacturer instructions. 

miRNA RT: miRs were polyadenylated using the Invitrogen NCode miRNA First-Strand cDNA Synthesis Kit (ThermoFisher, Waltham, MA, USA). The polyadenylated microRNA was reverse-transcribed to generate the first strand of cDNA per manufacturer’s instructions.

Preamplification and qPCR: Multiplex qPCR reactions used SYBR green and the ViiA 7 Real-Time PCR System (ThermoFisher, Waltham, MA, USA). The primers were custom-designed and synthesized by Integrated DNA Technologies (IDT, Coralville, IA, USA) [9]. The probe sets included primers for one of the five potential markers (*PSME2*, *NAMPT*, *APOA1*, *APOA4*, and *Hsa-Let-7g*) plus normalization and spike genes, so all three RNAs were run in the same well. One-µL RT samples were prepared for the preamplification mix reaction and underwent twelve cycles. Two µL of preamplification cDNA samples were diluted into 10µL PCR reaction mix, followed by RT PCR using SYBR Green Supermix (ThermoFisher, Waltham, MA, USA). Threshold cycles (Ct values) of qPCR reactions were extracted using QuantStudio™ Software V1.3 (Applied Biosystems, Foster City, CA, USA). Delta–delta CT method was used to calculate the expression of RNAs and then normalized.

### 2.3. Statistics

All statistical analyses were performed by one author (HSC) using SAS 9.4 (SAS Institute, Cary, NC, USA). Initial studies of laboratory tests are often modest in size, and metrics such as sensitivity, specificity, and negative/positive predictive values are highly influenced by disease prevalence. In contrast, the AUC is less affected by disease prevalence and provides an aggregate measure of performance across all classification thresholds and is one of the most useful parameters to evaluate a predictive model [11]. An AUC between 0.90 and 1.00 is considered excellent, one between 0.80 and 0.89 is good, 0.70–0.79 is fair, 0.60–0.69 is poor, and 0.50–0.59 is a failure [12]. We used AUC with detection rates (DRs) derived for three fixed false positive rates (FPRs), since an acceptable FPR could vary if the cost of misclassifications was part of model selection [13]. 

Gaussian modeling was not possible with the sample size, so a variety of linear regressions were tested with and without log base 2 transformation of expression and conversion to multiples of the median (MoM) [14]. The maternal characteristics available included maternal (MA) and gestational ages (GA), CRL, maternal weight (MW), race, and tobacco use. AUC was calculated for optimized models using a single threshold cut point and observed DRs determined for 3 fixed FPRs of 10, 20, and 30%. 

## 3. Results

The mean MA (±1SD) was 32.1 ± 1.6 y and the mean GA was 12.9 ± 0.1 weeks (12.0–14.1 weeks). Two controls were sampled at 14.0 and 14.1 weeks; the remaining 18 controls and 40 cases were sampled ≤13.9 weeks. The mean MW was 72.9 ± 15.5 kg. There were no differences between the case and control groups in either the maternal characteristics or the gestation at sampling. There was a trend toward obesity in cases (*p* = 0.09). Three women smoked tobacco: 1 control and 2 cases. The mean (±SD (range)) GA at delivery was 39.99 ± 0.8 weeks (39.1–41.8 weeks) for controls and 27.9 ± 2.3 weeks (24.1–31.8 weeks) for cases.

The five RNAs comprising the current panel were previously shown to have low expression in controls but overexpressed in cases [9,10], and in the current study, there were subjects in whom one or more of the RNA levels fell below assay detection, and more often than was seen in the 16–20 weeks study [10]. PSME2 was undetectable in 5% (1/20) of controls and 10% (4/40) of cases (Figure 1), NAMPT was undetectable in 60% (12/20) of controls and 52.5% (21/40) of cases, APOA1 was undetectable in 50% (10/20) of controls and 17.5% (7/40) of cases, APOA4 was undetectable in 65% (13/20) of controls and 32.5% (13/40) of cases, and, lastly, Hsa-LET-7g was undetectable in 10% (2/20) of controls and 5% (2/40) of cases. Overall, panel RNA expression was more likely to be undetected in controls than cases (Chi-square with Yates’ correction 6.2073, *p* = 0.012).

Two of the five panel RNAs, APOA1 and PSME2, were differentially expressed in women destined for sPTB ≤ 32 weeks (Figure 1). One subject in each group had expression levels below detection for both RNA markers (‘no calls’, 3.3% of subjects).

The predictive accuracy of the significant models is listed in Table 1. Converting gene expression to MoMs had a modest impact on the AUCs, while the addition of CRL and some maternal characteristics appeared to enhance predictive accuracy. APOA1 (MoMs) (AUC = 0.73, 95% CI 0.60–0.87) performed better than PSME2 (MoMs) (0.65, 95% CI 0.50–0.80). The combination of APOA1 and PSME2 did not significantly improve AUC over APOA1 alone. The two best models were: APOA1 (MoMs) + MA yielding an AUC = 0.79 (95% CI 0.66–0.91) and DR = 75% with a 30% FPR, and APOA1 (MoMs) + CRL, MA, MW, race, and tobacco use yielding an AUC = 0.79 (95% CI 0.66–0.91) and DR = 79% with a 30% FPR.

## 4. Discussion

No validated clinical screening test to date has provided even a ‘fair’ predictive accuracy (i.e., AUC > 0.70) for sPTB ≤ 32 weeks in the first trimester of parous, much less nulliparous, women. The current study is the third validation study of this five RNA panel specifically selected based on their relationship to a unique set of myometrial RNAs overexpressed in women delivering <32 weeks [9]. Like the prospective cohort validation study performed at 16–20 weeks [10], only two of the five panel RNAs were predictive, PSME2 and APOA1 at 12–14 weeks, and APOA1 was the best performer. The RNA-only AUC for *APOA1* = 0.73 (95% CI 0.60–0.87) provided a DR = 75%. This DR is similar to the RNA-only AUC = 0.76 (95% CI 0.65–0.87) achieved in a prospective cohort study at 16–20 weeks [10], suggesting the overall accuracy of the RNA panel for sPTB ≤ 32 weeks is unaffected by parity and may be useful in the first trimester. Combining *APOA1* with CRL, MW, MA, race, and tobacco use achieved an AUC = 0.79 (95% CI 0.66–0.91) for sPTB ≤ 32 weeks with a DR = 79%. This is similar to the AUC = 0.83 (95% CI 0.74–0.92) achieved in the 16–20-week samples with a DR = 77% when race and a history of prior PTB were combined [10]. The fact that the majority of women destined for sPTB ≤ 32 weeks can be identified by 12–14 weeks suggests that their risk is already ‘set’ in the absence of therapeutic intervention. It also suggests sPTB ≤ 32 weeks is not a syndrome with numerous causes that may begin in the second or third trimesters.

While medical history, CL, and fFN are used clinically to identify women at risk for sPTB, studies suggest the popularity of these tests is not based on their predictive accuracy, as all three have repeatedly been shown to be clinically poor tests [4,5,6]. Rather, caregivers and patients alike have nothing better to use. The risk prediction by maternal characteristics/history performs worse in nullipara compared to multipara, to the point of not being clinically useful for the prediction of sPTB > 32 weeks [15]. In the current study, the available maternal characteristics were limited but still seemingly associated with an AUC improvement. It is possible a larger sample size coupled with a more robust list of characteristics including prepregnancy-related hypertension, diabetes, or a family history of preterm birth would raise the first trimester AUC > 0.80 for sPTB ≤ 32 weeks. Dude [16] noted that a similar percentage of women with a short CL delivered prematurely, independent of parity, suggesting premature CL shortening is not impacted by prior pregnancy. Additionally, after prior studies of two different populations in which the five RNA panel at 16–20 weeks provided reproducible accuracy superior to conventional screening methods, this current study that was confined to first-trimester nullipara suggests plasma RNA testing has the potential to be a first-trimester screening tool.

Others have sought biomarkers for sPTB in asymptomatic women [17,18,19,20] but only Cook [21] and Hromadnikova [22] focused on the first trimester. Cook sampled women 12 to 21 weeks 6 d. There were 13 cases used for discovery but only 10 cases used for validation. Differentially expressed miRNA for sPTB < 34 weeks was sought using the Nanostring nCounter miRNA assay and Q rtPCR. In the 12–14 weeks 6 d group, Cook identified five differentially expressed miRNA. Another four miRNAs were differentially expressed at 15–21 weeks 6 d. In the validation phase, all 9 miRNAs were differentially expressed from the earliest gestation forward. The AUCs for individual miRNAs in the validation study ranged from 0.82 to 0.87 for the four best markers. This laboratory has not yet published a follow-up validation study. Hromadnikova [22] studied maternal peripheral leucocyte RNA but limited their investigation to 29 cardiovascular disease-associated microRNAs and sPTB < 37 weeks. Their best model predictive of sPTB < 37 weeks yielded an AUC = 0.81 that consisted of six miRNAs. The addition of MA and serum PAPP-A did not improve performance. The validation study has yet to be published.

There is reason to believe one or more RNAs comprising the current five RNA panel will be predictive of preeclampsia and/or EOP in the first trimester, just as occurred at 16–20 weeks [10]. Del Vecchio [23] described a first-trimester discovery study using RNAseq on plasma mRNA from 17 women in the late first trimester, 9 with a normal delivery at term, 5 with preeclampsia, and 3 with gestational hypertension. They identified 170 differentially expressed RNAs in the 8 women who developed pregnancy-related hypertension and selected 5 for logistic regression modeling on the same dataset used for discovery. The result was AUC = 0.86. One of their selected markers was *NAMPT*, which in the second trimester is a predictor of preeclampsia and EOP [10]. Their remaining PCF RNAs were *MMP8*, *SRPK1*, *S100A9*, and *S100A8*. No validation study has been published to date. In the current first-trimester validation study and in our prior second-trimester cohort validation study [10], the informative marker RNAs for sPTB ≤ 32 weeks achieved similar AUCs, but only *PSME2* was predictive of sPTB ≤ 32 weeks in both the first and second trimesters. Yet, in the first trimester, it did not add to the predictive accuracy of *APOA1* and may be a sample size issue. Not being able to detect the expression of *NAMPT* cannot be explained by a laboratory failure, as *APOA4* had the highest undetectable rates in the first trimester.

The clinical value of a screening test would be increased exponentially if there was an effective intervention available. Prophylactic progesterone is recommended in several scenarios. Additionally, while some professional organizations consider vaginal progesterone the standard of care for certain indications [24], there is clearly no international consensus on effectiveness [25,26]. Perhaps efficacy could be improved by a more accurate selection of at-risk women, as occurred with low-dose aspirin and EOP [27]. It was recently shown [28] that the rate of sPTB < 32 weeks in women deficient in DHA could be halved by DHA supplementation beginning in the early second trimester. Supplementation had no impact on sPTB rates in women with normal DHA levels. The ability to objectively identify a high-risk pool of women sPTB ≤ 32 weeks in the first trimester would facilitate future therapeutic trials, whether that be a current or new therapy. However, whether the prevention therapy is progesterone, DHA [28,29,30], cervical cerclage, or pessary, it is clear that new therapeutic approaches are needed. In addition, since the financial costs of sPTB ≤ 32 weeks represent the majority of healthcare costs in year one of life, even a small decrease in the rate of sPTB ≤ 32 weeks would likely be cost-effective [31].

As encouraging as the association of differentially expressed PCF RNAs is with pregnancy pathology, there is a clear need for large, real-world prospective studies to confirm the reproducibility of RNA markers among laboratories, and their predictive accuracy when other pathologies are present. 

Interestingly, *APOA1* at 16–20 weeks is a good predictor of EOP and not sPTB ≤ 32 weeks [10]. These differences are of clinical and biologic importance. First, they highlight the need for accurate dating. Second, they suggest these differentially expressed PCF RNAs that likely originate from the placenta [9] may alter their cell target as the placenta develops, perhaps by being packaged in a different transporter. The impact of gestation on informative RNAs for sPTB ≤ 32 weeks is consistent with our hypothesis as to why RNAs that are rigorously selected to predict sPTB ≤ 32 weeks could prove to be better predictors of EOP at other gestations [10]. In silica, the five PCF RNAs comprising the current panel have the potential to enhance intracellular calcium in smooth muscle, and in vitro, the overexpression of *APOA4* in immortalized human pregnant myometrial cells increases both intracellular calcium and cell contraction frequency [9]. This suggests that one of the potential effects of these overexpressed RNAs is interference with myometrial quiescence.

PCF RNAs rarely circulate naked; rather, they are associated with transporters such as extracellular vesicles, carrier proteins such as Argonaute 2 [32], or high-density lipoproteins [33]. These transporters slow RNA degradation and provide a targeting mechanism for RNA uptake by specific cells. sPTB and EOP share abnormal smooth muscle responsiveness. We have hypothesized sPTB occurs when marker RNAs are within transporters targeting myometrial smooth muscle, while EOP occurs when marker RNAs are within transporters targeting vascular smooth muscle (and/or endothelium) [10]. We have shown that increased *APOA1* is predictive of sPTB ≤ 32 weeks at 12 weeks, but also of EOP at 16–20 weeks. Thus, the maternal disease phenotype may depend as much on the RNA transporter as it does on the contained RNA. In support of this hypothesis, Yoffe et al. [34] described a first-trimester study of small noncoding RNAs using Next Generation Sequencing seeking differentially expressed RNAs that were predictive of EOP. They identified two of the same differentially expressed miRs that we identified in 2nd-trimester women [9], miR Let-7g and 99b. However, at 16–20 weeks, these were predictors of sPTB ≤ 32 weeks and not EOP, as Yoffe found at 11–13 weeks. 

The present validation study has several strengths. First, it is the third successful validation study of the five PCF RNA panel. Second, the samples were well documented and randomly chosen by an otherwise uninvolved party from a large biobank in a country previously not used for the testing of the panel. Third, the sample size, though limited, is similar to or larger than similar efforts from other laboratories. A fourth strength of the study is the use of a plasma RNA extraction method that increases total RNA yield from the nanograms extracted by existing commercial kits to microgram levels [35]. No PCF RNA test can be reproducible if the RNA extraction is not.

The study also has limitations. First, subject selection occurred a decade ago, early in our experience with the plasma transcriptome. We intentionally excluded women who either developed preeclampsia or had other medical comorbidities, out of concern comorbid conditions in a limited sample size could have significant but nonidentifiable effects. As a result, we have no information about the applicability of the five PCF RNA panel in the first trimester to preeclampsia. Second, the case-control nature of the study engenders a high prevalence of cases and creates the potential for a biased sample selection. The insulated sample selection process from a very large biobank should have minimized sample selection bias. Further, the use of AUC rather than sensitivity, specificity, etc., helps offset the impact of high disease prevalence. A third potential weakness is a reliance on regression analysis, as overfitting could exaggerate the AUCs. However, the changing expression of the marker RNAs as gestation advances prevented us from testing first-trimester expression with AUC curves generated from our larger second-trimester samples. Lastly, there was a limited number of maternal characteristics available for inclusion in the regression models. However, this particular weakness would have acted to lower the composite AUC rather than falsely elevate it. The inclusion of additional characteristics such as socioeconomic status, prepregnancy diseases (e.g., diabetes) and a family history of a prior preterm birth in the future may enhance accuracy [36,37].

## 5. Conclusions

Including the present study, there are now five validation studies [9,10,18,20] of varying size, conducted from 12.0 to 24 weeks, reaching similar conclusions: PCF RNAs combined with maternal characteristics identify with good accuracy those women who are destined for sPTB ≤ 32 weeks (labor ± PPROM), months in advance of the event. Expanded first-trimester testing will reveal whether the current markers are also useful for the prediction of other pregnancy disorders such as EOP/preeclampsia, as they were in the second trimester. This confirmation will make a first-trimester test available for the three most common causes of PTB ≤ 32 weeks and open the door to precision medicine in obstetrics (Figure 2). 

## 6. Patents

EU Patent No. 2646554 and US Patent No. 10,954,564.

## Figures and Tables

**Figure 1 biomedicines-11-01149-f001:**
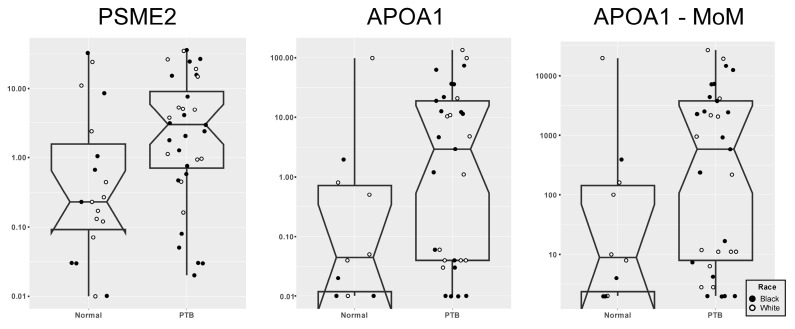
Expression Levels in Plasma Cell-Free *PSME2* and *APOA1*—cases and controls. Boxplots showing differentially expressed RNAs, e.g., *PSME2* (with maternal age), *APOA1*, and *APOA1* MoM. The boxes indicate the median and first and third quartiles, the lines demark the 10th and 90th percentile of the distributions, and the notches in the bars indicate median ±1.57 * interquartile range/Sqrt (n). The overlying circles indicate individual patient’s data, and these circles are coded by race. Nonoverlapping notches are “strong evidence” the medians differ. The folding of the box back towards the notch indicates a skewed distribution. Any sample outside whiskers might be an outlier. Footnote: Graphs created in R using ggplot2 library and the geom_boxplot() method (contact authors for the script).

**Figure 2 biomedicines-11-01149-f002:**
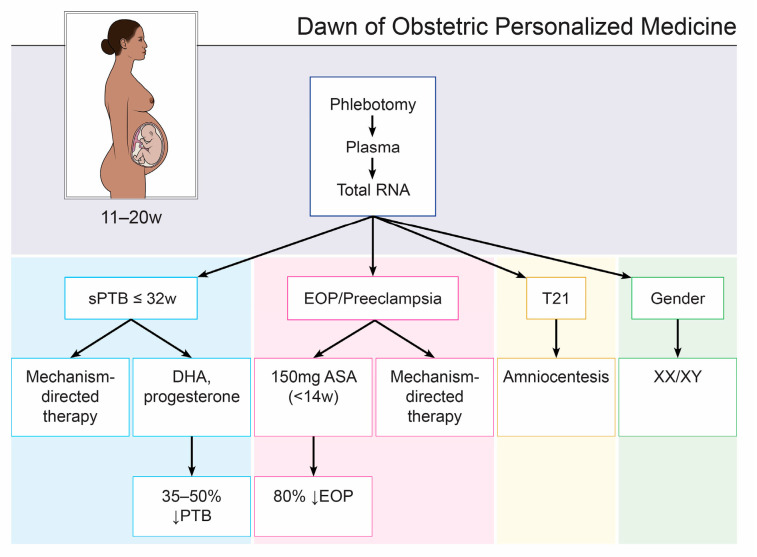
Dawn of Obstetric Personalized Medicine. Women may soon have the opportunity to choose screening based on a single plasma sample of cell-free RNA as early as the first trimester using a group of already validated tests and then initiate targeted therapy as desired.

**Table 1 biomedicines-11-01149-t001:** Performance of the Differentially Expressed RNAs in First Trimester Nullipara.

12-Week RNA Panel	AUC	*p* *	95% CI	Detection Rates of sPTB ≤ 32 Weeks		
				10% FPR	20% FPR	30% FPR
*PSME2*	0.65	0.05	0.50–0.80	18	35	63
*PSME2* (MoMs)	0.65	0.06	0.50–0.80	18	38	64
*PSME2* + MA	0.69	<0.02	0.54–0.83	26	45	55
*PSME2* (MoMs) + MA	0.67	<0.02	0.53–0.82	24	40	58
*PSME2* + CRL, MA, MW, race, tobacco	0.71	<0.005	0.57–0.85	32	39	58
*PSME2* (MoMs) + CRL, MA, MW, race, tobacco	0.70	<0.005	0.56–0.84	32	39	55
*APOA1*	0.72	<0.001	0.59–0.85	45	50	65
*APOA1* (MoMs)	0.73	<0.001	0.60–0.87	45	50	65
*APOA1* + MA	0.77	<0.0001	0.64–0.90	53	69	70
*APOA1* (MoMs) + MA	0.79	<0.0001	0.66–0.91	52	61	75
*APOA1* + CRL, weight, race, tobacco, MA	0.77	<0.0001	0.64–0.90	48	72	75
*APOA1* (MoMs) + CRL, MA, MW, race, tobacco	0.79	<0.0001	0.66–0.91	41	61	79
*PSME2* and *APOA1*	0.72	<0.002	0.58–0.86	41	50	59
*PSME2* and *APOA1* (MoMs)	0.73	<0.002	0.59–0.87	40	48	61
*PSME2* and *APOA1* + MA	0.78	<0.0001	0.65–0.90	50	64	74
*PSME2* and *APOA1* (MoMs) + MA	0.78	<0.0001	0.66–0.91	45	65	78
*PSME2* and *APOA1* + CRL, MA, MW, race, tobacco	0.78	<0.0001	0.65–0.90	48	66	72
*PSME2* and *APOA1* (MoMs) + CRL, MA, MW, race, tobacco	0.78	<0.0001	0.66–0.91	41	61	78

Key: AUC—area under the curve; FPR—fixed false positive rate; MA—maternal age; MW—maternal weight; CRL—crown–rump length; MoM—multiple of the median; tobacco—tobacco use. * Compared with 0.50.

## Data Availability

All data is available on request to the authors.

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
