# Peer review of "Maternal Plasma RNA in First Trimester Nullipara for the Prediction of Spontaneous Preterm Birth ≤ 32 Weeks: Validation Study"

_biomedicines, 2023, doi:10.3390/biomedicines11041149_

Round 1

Reviewer 1 Report

the manuscript is quite interesting but is superficially introduced and image quality is very low. In addition, important information on patients are not reported. The main flaws are the following: 

Introduction: a short introduction of PTB causes is missing. In fact,it deserves to be highlighted that the main cause of PTB are viral or bacterial infections (PMID: 26739007, 35943095). The presence of these infections significantly increase the inflammatory cytokines in the amniotic fluid weakening placental membranes favouring PPROM and causing PTB. 

Lines 52-75: It deserves to be pointed out that Giannubilo and colleagues evaluated the predictivity of serum HtrA1 levels in first trimester of pregnancy and found a good accuracy of PTB prediction (AUC=0.83) (PMID: 32102578).

Preamplification and qPCR:  primers sequence must be reported 

Figure 1: figure is  unreadable

A table showing the clinical and demographic characteristics of patients must be reported

An accurate revision of punctuation and syntax is recommended 

Reviewer 2 Report

A very interesting and valid study that shows how different tests can be used to achieve relative accuracy in the diagnosis of preterm labor. This study combines features that are already used in clinical practice with other genetic features. Some comments could improve the work:

1. Introduction. lines 52-54: not always nulliparous women are at higher risk of preterm delivery.

Why these RNAs (line 68), and not others, were chosen for early diagnosis of preterm delivery? what do they encode or regulate?

2. Material and methods. An ethics committee should be reported with a reference to the site where the participants were recruited or, at least, the institution where the maximum of the research was performed.

It should be specified how and from where the maternal characteristics variables were collected (line 125).

In the statistical section, it is not clear neither the program used, nor the probability of error that is taken as significant, nor is it specified how the descriptive statistics of the data were shown. 

3. Results.  Lines 141-146: It is more intuitive to report percentages.

The figure 1 is of poor quality, it is very difficult to see it. This figure caption should be completely rewritten, it is not necessary to tell what the IQRs are, rather, what the figure itself describes, the overlapping points, etc. 

4. Discussion. Although it is fairly well narrated, I think it is missing a description of what characteristics women should meet to be tested by RNAs, since this technique is expensive and not all OBS services may have it available.

In Figure 2, it is unrepresentative of the work, where would the detemrination of RNAs be implemented? What orte points would be taken at the levels of these transcripts?

Minor comments:

- In the conclusions, you should specify concrete data of the work, RNAs described, which is the best candidate, level, in which characteristics. 

Round 2

Reviewer 1 Report

the manuscipt can be accepted in the present form 

Author Response

The authors thank the Reviewer for their criticisms and recommendations.  While we were pleased they had no further requests, we have again made additional  changes to improve readability.

Reviewer 2 Report

Thanks to the authors for this new version. Although I consider that more details should be given to the reader (without duplicating data) to all the information would be available. I understand that this article is part of a larger set of publications which has already been cited. Perhaps, in the figure caption 2, the authors' response could be included as it seems clearer. I have no further considerations

Author Response

(The authors gave the same response as above.)
